# Nanophotosensitizers for Folate Receptor-Targeted and Redox-Sensitive Delivery of Chlorin E6 against Cancer Cells

**DOI:** 10.3390/ma13122810

**Published:** 2020-06-22

**Authors:** Min-Suk Kook, Chang-Min Lee, Young-Il Jeong, Byung-Hoon Kim

**Affiliations:** 1Department of Oral and Maxillofacial Surgery, School of Dentistry, Chonnam National University, Gwangju 61186, Korea; omskook@chonnam.ac.kr; 2Department of Dental Materials, School of Dentistry, Chosun University, Gwangju 61452, Korea; ckdals1924@daum.net; 3Research Institute for Convergence of Biomedical Science and Technology, Pusan National University Yangsan Hospital, Gyeongnam 50612, Korea

**Keywords:** oral cancer, photodynamic therapy, chlorin e6, nanophotosensitizers

## Abstract

In this study, FA–PEG_3500_-ss-Ce6tri copolymer was synthesized to deliver photosensitizers via redox-sensitive and folate receptor-specific manner. Folic acid (FA) was attached to amine end of poly (ethylene glycol) (PEG_3500_) (FA–PEG_3500_ conjugates) and cystamine-conjugated chlorin e6 (Ce6) (Ce6-cystamine conjugates). FA–PEG_3500_ was further conjugated with Ce6-cystamine to produce FA–PEG_3500_-ss-Ce6 conjugates. To the remaining amine end group of Ce6-cystamine conjugates, Ce6 was attached to produce FA–PEG_3500_-ss-Ce6tri. Nanophotosensitizers of FA–PEG_3500_-ss-Ce6tri copolymer were smaller than 200 nm. Their shapes were disintegrated by treatment with GSH and then Ce6 released by GSH-dependent manner. Compared to Ce6 alone, FA–PEG_3500_-ss-Ce6tri copolymer nanophotosensitizers recorded higher Ce6 uptake ratio, reactive oxygen species (ROS) production and cellular cytotoxicity against KB and YD-38 cells. The in vitro and in vivo study approved that delivery of nanophotosensitizers is achieved by folate receptor-sensitive manner. These results indicated that FA–PEG_3500_-ss-Ce6tri copolymer nanophotosensitizers are superior candidate for treatment of oral cancer.

## 1. Introduction

The incidence rate and mortality of oral cancer has been continuously increasing worldwide in the last two decades. Oral cancer patients are known to have the lowest survival rate at the five-year mark among major cancer types [1]. Oral cancers are frequently diagnosed in their advanced stages and when they are metastasized to other regions—despite the easy accessibility for identification of a cancer lesion [2,3]. Even though surgery, chemotherapy, radiotherapy and immunotherapy are currently considered as treatments for oral cancers, high recurrence rates and metastasis to distant regions after sole or combined treatments are still problematic and resulted in low survival rates [4,5,6,7]. Therefore, a novel treatment regimen for oral cancers is required to improve therapeutic efficacy with reduced side effects.

Photodynamic therapy (PDT) is believed to be a safe treatment option for malignant disorders because it consists of light, oxygen species and photosensitizer [8,9]. PDT treatment for cancer can minimize adverse side effects because photosensitizers produce an excess amount of reactive oxygen species (ROS) under the irradiation condition of specific wavelength of light and have negligible cytotoxicity against surrounding normal tissues and cells. This means cells only in the irradiated field can be eliminated [8,9,10]. Many researchers and clinicians have been trying to eradicate malignant or premalignant tumors using PDT [9,10,11,12,13,14,15]. It has been speculated that PDT is a suitable treatment option for oral cancers because oral cancers including gingival cancer occur in oral cavity, of which can be easily accessed via visible light for PDT treatment, and, from these point of view, it has been used in control of early or advanced stages of oral cancers [15,16,17,18,19]. Nevertheless, Nauta et al. report that unsatisfactory long-term results were observed in PDT on squamous cell carcinoma using porfimer sodium and PDT efficacy was dependent on the wavelength of light [17]. Furthermore, it is known that the drug-resistant problem in oral cancer cells also occurs in PDT with 5-aminolevulinic acid (5-ALA) [20]. Traditional photosensitizers such as 5-ALA also have limited tumor specificity followed by the problem of them spreading throughout the body [10,21,22]. This is because, even though photosensitizer itself is activated under the specific wavelength of light, phototoxicity can be provoked, so it is required to block the sun light [22]. Photosensitizers should be concentrated in the diseased tissues or specific cells to minimize these problems.

Polymeric- or nano-photosensitizers have been investigated to solve problems of traditional photosensitizers since nanoparticles have been used as a carrier for targeted delivery of anticancer drug against specific region of human body [23,24,25,26]. Nanocarriers emphasize anticancer activity of traditional anticancer drugs and can be used for diagnostic/therapeutic purposes [23,24]. In particular, nanoparticles decorated with targeting molecules on the surface have great potential to target cancer cells because cancer cells express many kinds of molecular receptors excessively such as CD44 and folate receptors [25,26,27,28,29,30,31,32]. For example, hyaluronic acid (HA) can recognize CD44 receptors on the cancer cells and then they can be used to detect CD44-overexpressing cancer cells through conjugation with Ce6 [25,26]. Lee and Jeong reported CD44 receptor-sensitive delivery of HA-Ce6-poly (L-histidine) copolymer nanoparticles against CD44-overexpressing cancer cells and their suppressive effect of tumor [25]. Furthermore, folate receptors on the cell surface also can be applied to target cancer cells since expression of folate receptors is known to be increased according to the malignancies of tumor [27,28,29]. Folic acid (FA)-conjugated polymer nanoparticles concentrated more photosensitizers or anticancer drugs into tumor tissue through the folate-receptor-mediated pathway compared to the normal tissue [30,31,32]. Folate receptor-mediated drug targeting using nanoparticles may solve drawbacks of traditional photosensitizers.

For this study, we synthesized FA-conjugated polymeric nanophotosensitizers composed of FA, PEG_3500_ and Ce6 for targeted PDT of oral cancer. FA was attached to one end of PEG and then the other end of PEG_3500_ was conjugated with Ce6 trimer using disulfide linkage for folate-receptor-specific and redox-potential-triggering delivery of Ce6 (FA–PEG_3500_-ss-Ce6tri). Physicochemical properties and PDT efficacy of FA–PEG_3500_-ss-Ce6tri were studied with oral cancer cell lines.

## 2. Materials and Method

### 2.1. Materials

PEG 2-aminoethyl ether acetic acid (NH_2_-PEG_3500_-COOH, Mn = 3500 g/mol) was obtained from Sigma-Aldrich (St. Louis, MO, USA). Chlorin e6 was obtained from Frontier Scientific, Inc. (Logan, UT, USA). Cystamine dihydrochloride, L-glutathione reduced (GSH), 1,4-dithiothreitol (DTT), *N*-hydroxysuccinimide (NHS), *N*-(3-dimethylaminopropyl)-N’-ethylcarbodiimide hydrochloride (EDAC), triethylamine (TEA), 3-(4,5-dimethylthiazol-2-yl)-2,5-diphenyltetrazolium bromide (MTT) and dimethyl sulfoxide (DMSO) were obtained from Sigma-Aldrich (St. Louis, MO, USA). The dialysis membrane (Molecular weight cutoff (MWCO) size: 1000 and 2000 g/mol) was obtained from Spectrum/Por Lab., Inc. (Los Angeles, CA, USA).

### 2.2. Synthesis of FA–PEG_3500_-ss-Ce6tri Copolymer

Synthesis of FA–PEG_3500_ conjugates: FA (44.1 mg) dissolved in 10 mL DMSO was mixed with equivalent mole of EDAC and NHS followed by magnetic stirring for 3 h. Then, NH_2_-PEG_3500_-COOH (350 mg) was added and then stirred magnetically. Twelve hours later, reactants were put into dialysis tube (MWCO: 2000 g/mol). These were dialyzed against distilled water for 1 day with exchange of water at every 2–3 h intervals. Following this, dialyzed solution was freeze-dried.

Synthesis of Ce6-cystamine conjugates: Ce6 (59.7 mg) in DMSO (10 mL) was mixed with three equivalent amounts of EDAC and NHS followed by magnetic stirring for 6 h. To this solution, 337.8 mg (fifteen equivalents vs. Ce6) cystamine dihydrochloride was added with few amount of TEA and then stirred magnetically for 24 h. Resulting solution was dialyzed (MWCO of dialysis membrane: 1000 g/mol) using distilled water for 1 day with exchange of water every 3 h intervals. Ce6 functionalized with three cystamine moieties (Ce6-(-NSSNH_2_)_3_) as a black solid was obtained by freeze-drying for 2 days.

Synthesis of FA–PEG_3500_-ss-Ce6tri copolymer: FA–PEG_3500_ conjugates (392 mg) in 10 mL of DMSO were mixed with equivalent mole of EDAC and NHS. Thirty minutes later, Ce6-(-NSSNH_2_)_3_ (100 mg) was added and then further stirred for 24 h to obtain FA–PEG_3500_-ss-Ce6 copolymer. Ce6 was separately reacted with equivalent mole of EDAC/NHS system for 6 h to obtain NHS-activated Ce6. Two molar equivalents of NHS-activated Ce6 were mixed with reaction of FA–PEG_3500_-ss-Ce6 copolymer solution followed by magnetic stirring 24 h. Resulting solution was dialyzed (MWCO of dialysis membrane: 2000 g/mol) against distilled water for 2 days and then freeze-dried for 2 days.

### 2.3. Characterization of Copolymer

The structure of synthesized conjugates was elucidated by ^1^H nuclear magnetic resonance (NMR) spectra (Varian Unity Inova 500 MHz NB High Resolution Fourier Transform NMR (FT NMR); Varian, Inc, Santa Clara, CA, USA).

### 2.4. Preparation of FA–PEG_3500_-ss-Ce6tri Nanophotosensitizers

To form nanoparticles, FA-PEG_3500_-ss-Ce6tri (10 mg) in 2 mL DMSO was poured into deionized water and then put into dialysis tube (WCO = 2000 g/mol). This was dialyzed against deionized water for 1 day and deionized water was exchanged 2–3 h intervals. Dialyzed aqueous solution was used to analyze or evaluate PDT efficacy using cancer cells.

Ce6 content measurement: To analyze Ce6 contents in the copolymer, FA–PEG_3500_-ss-Ce6 or FA–PEG_3500_-ss-Ce6tri copolymer (5 mg) in 50 mL phosphate-buffered saline (PBS) (pH 7.4, 0.01 M, 20 mM DTT) was magnetically stirred for 24 h and then diluted with DMSO ten times. Ce6 contents were measured with an Infinite M200 pro microplate reader (Tecan, Mannedorf, Switzerland) (Excitation wavelength: 407 nm, emission wavelength: 664 nm). For standard test, was diluted 20 times with PBS (pH 7.4, 0.01 M, 20-mM DTT) and then diluted ten times with DMSO. Ce6 content, % (*w/w*) = (Ce6 weight/weight of FA–PEG_3500_-ss-Ce6tri copolymer) × 100.

### 2.5. Nanophotosensitizer Characterization

Morphology of nanophotosensitizers: transmission electron microscope (TEM; H-7600, Hitachi Instruments, Ltd., Tokyo, Japan) was employed. Aqueous solution of nanophotosensitizers was placed onto a carbon-film-coated grid and then was dried at room temperature for 6 h. The accelerating voltage for observation of nanophotosensitizers was 80 kV.

Size distribution of nanophotosensitizers: Zetasizer (Nano-ZS, Malvern, Worcestershire, UK) was employed to measure particle size (nanophotosensitizer solution, 0.1% (*w/v*).

Fluorescence spectra of nanophotosensitizers: Nanophotosensitizer solution were scanned between 500 nm and 800 nm (excitation wavelength: 400 nm) using Infinite M200 pro microplate reader.

Fluorescence image of nanophotosensitizers: Nanophotosensitizer solution was observed with Maestro 2 small animal imaging instrument (Cambridge Research and Instrumentation, Inc., Boston, MA, USA). Nanophotosensitizer solution (0.1-mg/mL as a FA–PEG_3500_-ss-Ce6tri copolymer) in PBS (pH 7.4, 0.01 M) was incubated at 37 °C for 3 h with or without GSH.

### 2.6. Ce6 Release Study

Release behavior of Ce6 was carried out in PBS (0.01 M, pH 7.4, GSH 20 mM). Five milliliters of nanophotosensitizer aqueous solution (1 mg/mL) was put into dialysis membrane and dialysis membrane was immersed into 45 mL PBS (pH 7.4, 0.01 M, GSH 20 mM). Release of Ce6 was carried out at 37 °C with gentle shaking (100 rpm). PBS was replaced with fresh PBS to avoid saturation. Ce6 release was evaluated using an Infinite M200 pro microplate reader (excitation wavelength: 407 nm, emission wavelength: 664 nm). All experiments were triplicated to calculate mean ± standard deviation (S.D.).

### 2.7. Cell Culture

KB human epithelial carcinoma and YD-38 squamous carcinoma cells were purchased from the Korean Cell Line Bank Co. (Seoul, Korea). Cells were cultured with Roswell Park Memorial Institute (RPMI)-1640 medium supplemented with 10% (*v/v*) fetal bovine serum and 1% (*v/v*) antibiotics in a 5% CO_2_ incubator (37 °C).

### 2.8. PDT Effect In Vitro

KB or YD-38 cells (2 × 10^4^ cells/well) in 96 well plates were treated with FA–PEG_3500_-ss-Ce6tri nanophotosensitizers for 2 h in CO_2_ incubator (37 °C). For comparison, Ce6 in DMSO was diluted 20 times with Fetal bovine serum (FBS)-free (Final concentration of DMSO: <0.5% (*v/v*)). For nanophotosensitizer treatment, aqueous solution of nanophotosensitizers was sterilized with 1.2 μm syringe filter and then diluted with FBS-free media. Two hours later, cells were washed with PBS twice, serum-free media (100 µL) was added, and then irradiated at 664 nm using expanded homogenous beam (SH Systems, Gwangju, Korea). The dose of irradiated light was measured with a photo radiometer (Delta Ohm, Padua, Italy) (Irradiation time: 9 min 29 s (2.0 J/cm^2^)). After that, cells were cultured 1 day with serum-free media at 37 °C (5% CO_2_). MTT proliferation assay was employed to evaluate the viability of cancer cells. Thirty micrometers MTT solution (5-mg/mL in PBS) was added to each wells and then cells were further incubated for 3 h at 37 °C (5% CO_2_). Supernatants were removed and, after that, 100 µL DMSO was added. Absorbance was measured at 570 nm using an Infinite M200 pro microplate reader. The results were calculated from eight wells and expressed as mean ± S.D. All procedure was carried out in dark condition.

Cancer cells were not irradiated to evaluate dark toxicity of nanophotosensitizer and following procedures were similar to described above.

### 2.9. Relative Ce6 Uptake Comparison

KB cells or YD-38 cells (2 × 10^4^ cells/well) in 96-well plate were treated with Ce6 or nanophotosensitizers. Two hours later, cells were washed with PBS (0.01 M, pH 7.4) twice and then lysed with 50 µL of lysis buffer (GenDEPOT, Barker, TX, USA). The intracellular Ce6 ratio was measured with Infinite M200 pro microplate reader (Tecan Trading AG, Männedorf, Switzerland) (Excitation wavelength: 407 nm, emission wavelength: 664 nm).

### 2.10. Fluorescence Microscopy for Observation of Cells

5 × 10^5^ cells seeded on the cover glass in 6 well plates were exposed to Ce6 or nanophotosensitizers. Ninety minutes later, cells were washed with PBS twice and fixed with 4% paraformaldehyde solution. They were mounted with immobilization solution (Immu-Mount, Thermo Electron Co., Pittsburgh, PA, USA) to observe cell morphology using fluorescence microscope (Eclipse 80i; Nikon, Tokyo, Japan).

### 2.11. Folate Receptor-Targeting of KB Cells

KB cells (5 × 10^5^ cells/well) seeded onto cover glass in 6-well plates were cultured overnight at 37 °C in a 5% CO_2_ incubator. Prior to nanophotosensitizer treatment, cells were treated with 5-mM folic acid for 1 h to study folate receptor sensitivity. Cells were washed with PBS and then exposed to Ce6 or nanophotosensitizers (2-μg/mL as a Ce6 concentration) for 1 h. Following this, cells washed with PBS was fixed with 4% paraformaldehyde solution and then mounted with Immu-Mount solution (Thermo Electron Co, Pittsburgh, PA, USA) to observe cells using fluorescence microscope.

### 2.12. ROS Production

Cancer cells (2 × 10^4^) were treated with Ce6 or nanophotosensitizer for 2 h at 37 °C. To evaluate ROS production from cells, DCFH-DA (final concentration: 20 µM) was also added. Two hours later, cells were washed with PBS twice and then replaced with 100 µL fresh phenol red free RPMI media. The irradiation of cells was performed for 9 min 29 s (2.0 J/cm^2^) at 664 nm using expanded homogenous beam. Intracellular ROS generation was analyzed with microplate reader (Infinite M200 pro microplate reader (Tecan), Excitation wavelength, 485 nm; emission wavelength, 535 nm).

### 2.13. In Vivo Fluorescence Imaging Study Using Pulmonary Metastasis Model of KB Cells

KB cells (1 × 10^6^ cells) were intravenously (i.v.) injected via tail vein of nude BALb/C mouse (5 weeks, 20 g). Three weeks later, nanophotosensitizers (dose: 10 mg/kg, injection volume: 200 μL) were administered intravenously (i.v.) via the tail vein of the mouse. Prior to administer nanophotosensitizer, folic acid in PBS (dose: 20 mg/kg) was i.v. administered 1 h before to study folate receptor sensitivity. Twenty-four hours later, the mice were sacrificed to obtain each organ. Fluorescence images of each organ were observed with Mestro^TM^ 2 small animal imaging instrument (Cambridge Research and Instruments, Inc., Woburn, MA, USA).

All animal study was carried out under the guidelines of the Pusan National University Institutional Animal Care and Use Committee (PNUIACUC). The protocol of animal study was reviewed and strictly monitored by the PNUIACUC on their ethical procedures and scientific care and was approved (Approval Number: PNU-2017–1610).

### 2.14. Statistical Analysis

The statistical significance was calculated with Student’s t test using SigmaPlot^®^ program (SigmaPlot^®^ v.11.0, Systat Software, Inc., San Jose, CA, USA) and evaluated as *p* < 0.05 as the minimal level of significance.

## 3. Results

### 3.1. Synthesis of FA–PEG-ss-Ce6tri Copolymer

To synthesize FA–PEG-ss-Ce6tri copolymer, FA–PEG conjugates and Ce6-cystamine conjugates were prepared as shown in Figure 1. FA was attached to the amine end of bifunctional PEG (NH_2_-PEG_3500_-COOH) using EDAC/NHS system to obtain FA-conjugated PEG_3500_. Specific peaks of PEG_3500_ and FA were displayed at 3.5 ppm and 2.0–9.4 ppm, respectively (data not shown). Otherwise, excessive amount of cystamine was added to Ce6 solution to attach cystamine against three carboxylic acid of Ce6. To synthesize Ce6-cystamine conjugates, 3 equivalents of EDAC/NHS system were added to activate carboxylic acid of Ce6 and then cystamine was attached to NHS-activated Ce6. The ethylene protons of cystamine were confirmed at about 2.8 ppm while methylene protons and (–CH=CH_2_) protons of Ce6 were confirmed at 1.2–1.8 ppm and 6.0–7.0 ppm, respectively (Data not shown). FA–PEG_3500_ conjugates were attached to the one of amine end of Ce6-cystamine conjugates to produce FA–PEG_3500_-ss-Ce6 copolymer. Furthermore, two equivalents of Ce6 were activated with EDAC/NHS (NHS-activated Ce6) to synthesize FA–PEG_3500_-ss-Ce6tri copolymer, i.e., two equivalents of NHS-activated Ce6 were attached to the remaining two amine ends of Ce6-cystaime of the FA–PEG_3500_-ss-Ce6 copolymer to produce FA–PEG_3500_-ss-Ce6tri copolymer. Figure 1b shows that specific peaks of PEG, FA, cystamine and Ce6 were observed from ^1^H NMR spectra of FA–PEG_3500_-ss-Ce6tri copolymer, indicating that copolymer was successfully synthesized. As shown in Table 1, Ce6 contents in the FA–PEG_3500_-ss-Ce6 and FA–PEG_3500_-ss-Ce6tri copolymer were 11.8% (*w/w*) and 28.3% (*w/w*), respectively. Experimental contents of Ce6 were almost similar to the theoretical value in the copolymer, indicating that Ce6 trimer must be formed in the copolymer structure because free Ce6 and small molecules were removed by dialysis procedure. Particle sizes of the FA–PEG_3500_-ss-Ce6 and FA–PEG_3500_-ss-Ce6tri copolymer were 96.2 ± 7.3 nm and 189.1 ± 10.8 nm, respectively.

### 3.2. Characterization of FA–PEG_3500_-ss-Ce6tri Copolymer Nanophotosensitizers

Nanophotosensitizers of FA–PEG_3500_-ss-Ce6tri copolymer were prepared by dialysis procedure. Since FA–PEG segment and Ce6tri segment has hydrophilic and lipophilic characters, FA–PEG_3500_-ss-Ce6tri copolymer nanophotosensitizers have core-shell structures, i.e., Ce6tri segment consisting inner-core of the nanophotosensitizers while FA–PEG segment forming outer-shell. Nanoparticles from FA–PEG-ss-Ce6 copolymer have small particle size less than 100 nm while FA–PEG_3500_-ss-Ce6tri copolymer nanophotosensitizers have increased particle size such as 189.1 nm as shown in Figure 2A(a) and Table 1. Furthermore, they formed spherical nanoparticles having small diameter of around 200 nm as shown in Figure 2B(a). Since nanophotosensitizers of FA–PEG_3500_-ss-Ce6tri copolymer have disulfide bond for redox sensitivity, they were reacted with GSH as shown in Figure 2 and Figure 3. When nanophotosensitizers were treated with GSH, size distribution of nanophotosensitizers showed multimodal patterns while absence of GSH showed monomodal distribution patterns (Figure 2A(b–d)). Furthermore, morphology of nanophotosensitizers became irregular form by treatment with GSH as shown in Figure 2B(b)) while no treatment with GSH resulted in spherical morphology (Figure 2B(a)). These results indicate that nanophotosensitizers can be disintegrated by GSH (Figure 2B(b)).

Figure 3a shows the GSH effect on the fluorescence spectra of FA–PEG_3500_-ss-Ce6tri copolymer nanophotosensitizers. Fluorescence intensity was gradually increased in the course of GSH concentration. This result indicated that Ce6 was liberated by redox-sensitive pathway, i.e., nanophotosensitizers have GSH-sensitivity. In the aqueous solution, released Ce6 became activated state while Ce6 in the nanophotosensitizers are existed as a ground state with decreased fluorescence intensity (Figure 3a). Figure 3b shows that GSH in the release medium accelerated Ce6 release rate while absence of GSH induced slower release rate of Ce6. These results approved that nanophotosensitizers have redox sensitivity.

### 3.3. Biologic PDT Effect of Nanophotosensitizers against Cancer Cells

Anticancer activity of PDT nanophotosensitizers of FA–PEG_3500_-ss-Ce6tri copolymer was studied with KB and YD-38 as shown in Figure 4, Figure 5 and Figure 6. Figure 4a,b shows that nanophotosensitizers revealed significantly higher Ce6 uptake efficacy than that of Ce6 alone. Ce6 uptake ratio was gradually increased in the course of nanophotosensitizer concentration both KB and YD-38 cells. Fluorescence microscopy also approved that fluorescence intensity was stronger in nanophotosensitizer treatment than that of Ce6 alone.

Similar to Ce6, nanophotosensitizers have little toxicity in dark conditions i.e., viability of cancer cells maintained higher than 80% until 5 μg/mL of nanophotosensitizers in both Ce6 alone and nanophotosensitizers, indicating that nanophotosensitizers have negligible intrinsic cytotoxicity against cancer cells (Figure 5a). However, Figure 5b shows that nanophotosensitizers have higher PDT-induced cytotoxicity than that of Ce6 alone against KB and YD-38 cells. That means viability of KB cells was less than 40% at 1 μg Ce6/mL of nanophotosensitizers while Ce6 alone showed similar PDT efficacy at 5 μg/mL. As shown in Table 2, IC_50_ values of nanophotosensitizers were significantly lower than that of Ce6, i.e., nanophotosensitizers have superiority for PDT of cancer cells. Furthermore, ROS production of nanophotosensitizers was also significantly higher than that of Ce6 alone both in KB cells and YD-38 cells as shown in Figure 5c. These results indicate that nanophotosensitizers have superior PDT efficacy compared to Ce6 alone.

Figure 6 showed the folate receptor targetability of FA–PEG_3500_-ss-Ce6tri nanophotosensitizers. To investigate sensitivity of nanophotosensitizers against folate receptor, KB cells were pretreated with FA prior to treatment with nanophotosensitizers to block folate receptor (KB cells) and then nanophotosensitizers were treated to cells. Figure 6a shows that KB cells without blocking of folate receptor (FA(-)) revealed stronger fluorescence intensity than that with blocking of folate receptor. However, Ce6 alone did not significantly change fluorescence intensity of cells with or without blocking of folate receptor. These results approve that FA–PEG_3500_-ss-Ce6tri nanophotosensitizers have folate receptor-sensitivity. Furthermore, pretreatment with free FA also decreased Ce6 uptake ratio and ROS production in KB cells (Figure 6b,c) while the FA pretreatment with free FA did not affect the Ce6 uptake or ROS production in the treatment of Ce6 alone. PDT efficacy also responded to the FA receptor blocking (Figure 6d). These results approve that FA–PEG_3500_-ss-Ce6tri nanophotosensitizers can be delivered by the folate receptor sensitive manner in the cancer cells.

Figure 7 shows that folate receptor targetability of FA–PEG_3500_-ss-Ce6tri nanophotosensitizers was also studied using pulmonary metastasis model of KB cells. KB cells were i.v. administered via the tail vein of the mice to induce pulmonary metastasis. To test folate receptor sensitivity, FA was i.v. administered via the tail vein of the mice to block folate receptor of KB tumor. Liver and lung revealed stronger fluorescence intensity than that in the other organs in the absence of FA pre-administration (FA(-)). When FA was pretreated to the mouse (FA(+)), fluorescence intensity in the lung significantly decreased compared non-treatment (FA(-)) model. These results also approved that FA–PEG_3500_-ss-Ce6tri nanophotosensitizers have folate-receptor targetability.

## 4. Discussion

Oral cancer such as gingival cancers is difficult to cure because traditional regimens for treatment have limited efficacy and difficulties for diagnosis [2,3]. Even though surgery, chemotherapy, radiotherapy and immunotherapy are currently considered as the treatment for the oral cancer, high recurrence rate and metastasis to the distant regions after sole or combined treatment are still problematic and resulted in low survival rates [4,5,6,7]. From this point of view, PDT regimen is believed to be ideal candidate to improve life quality. However, drawbacks of PDT such as sun–shade problem limit its clinical application [22]. If photosensitizers can be preferentially delivered to tumors rather than surrounding normal tissues, its fluorescence properties can be used to detect and treat tumors specifically [8]. Actually, we showed that nanophotosensitizers can be specifically delivered to tumor cells via folate receptor-mediated manner as shown in Figure 6 and Figure 7. That is, FA–PEG_3500_-ss-Ce6tri nanophotosensitizers were delivered preferentially through folate receptor specific manner against KB cells (Figure 6). In vivo pulmonary metastasis model of KB cells also proved that FA–PEG_3500_-ss-Ce6tri nanophotosensitizers can be delivered and concentrated in the tumor cells by folate receptor specific manner. These results mean that photosensitizers can be specifically delivered to the cancer cells via specific receptors of oral cancer cells and then concentrated in the tumor tissues rather than normal counterpart. Li et al. also reported that photosensitizer conjugated with FA–PEG can be delivered by folate receptor-specific manner against folate-receptor-overexpressing cancer cells [32]. In particular, folate receptor expression is known to be increased in the patients having periodontal disease [33]. The expression values of folate-receptor 1 are known to be elevated in gingivitis and periodontitis groups than those of healthy group [33].

Meanwhile, reduction-oxidation (redox)-potential is also known to be elevated in the patients having periodontal disease [34,35]. Wong et al. reported that GSH level in oral cancer tissues was significantly higher than that of adjacent non-tumor tissue parts and its level has co-relationship with tumor size [35]. We showed that FA–PEG_3500_-ss-Ce6tri nanophotosensitizers were specifically disintegrated by GSH-dependent manner and then Ce6 can be liberated from nanophotosensitizers by response to the concentration of GSH (Figure 3). Since intracellular level of GSH in cancer cells is higher than that in extracellular level, GSH-dependent liberation of Ce6 may be considered for intracellular targeting of photosensitizers [36,37]. The fact that tumor cells have higher redox-potential that adjacent normal cells and tissues can be also used to detect tumor [38,39]. Even though FA–PEG-photosensitizer conjugates reported by Li et al. also have a targeting potential against cancer cells, their conjugates have no redox-sensitive linkages and then simply provide folate-receptor sensitivity [32]. In our system, Ce6 domain in the FA–PEG_3500_-ss-Ce6tri nanophotosensitizers is existed as a timer, i.e., Ce6 molecules are linked with each other via disulfide linkages. These properties of FA–PEG_3500_-ss-Ce6tri copolymer may provide superior redox-sensitivity of nanophotosensitizers as shown in Figure 3.

PDT may offer new approaches for detection and treatment of oral cancers in the early stage. Mallia et al. used ALA-induced normalized fluorescence to detect anatomic locations of the oral cavity and to improve the diagnostic contrast/accuracy of oral cancers [38]. In addition to that, PDT can be used to non-invasive treatment of precancerous lesions. Maloth et al. reported that positive response was obtained from PDT approach against oral leukoplakia lesions oral lichen planus lesions [39]. Since traditional photosensitizers have low tumor-specificity and also disperse to the normal cells or tissues, however, tumor-specific delivery of photosensitizers may offer improved diagnosis and treatment regimen of oral cancers. FA–PEG_3500_-ss-Ce6tri nanophotosensitizers have redox-sensitive and folate-receptor sensitive behavior against oral cancer cells as shown in Figure 3, Figure 6 and Figure 7. These results may enable us to detect and treat oral cancer cells with minimal side-effects against normal cells/tissues. In particular, tumor tissue-specific cytotoxicity during PDT can be triggered by irradiation of specific disease sites even though photosensitizers are fully distributed around the human body because PDT with did not reveal cellular cytotoxicity in the absence of light irradiation [40].

In conclusion, we synthesized FA–PEG_3500_-ss-Ce6tri copolymer to deliver photosensitizers against cancers with redox-sensitive and folate-specific manner. FA–PEG_3500_-ss-Ce6tri copolymer nanophotosensitizers have reduced particle sizes (<200 nm) and spherical shapes. They were disintegrated and Ce6 released by GSH-dependent manner. Higher Ce6 uptake ratio, ROS production and cellular cytotoxicity against KB and YD-38 cells were obtained by treatment of FA–PEG_3500_-ss-Ce6tri copolymer nanophotosensitizers. Furthermore, we approved folate receptor-specificity of nanophotosensitizers in vitro and in vivo. We suggest that FA–PEG_3500_-ss-Ce6tri copolymer nanophotosensitizers are the promising candidate for PDT of oral cancers.

## Figures and Tables

**Figure 1 materials-13-02810-f001:**
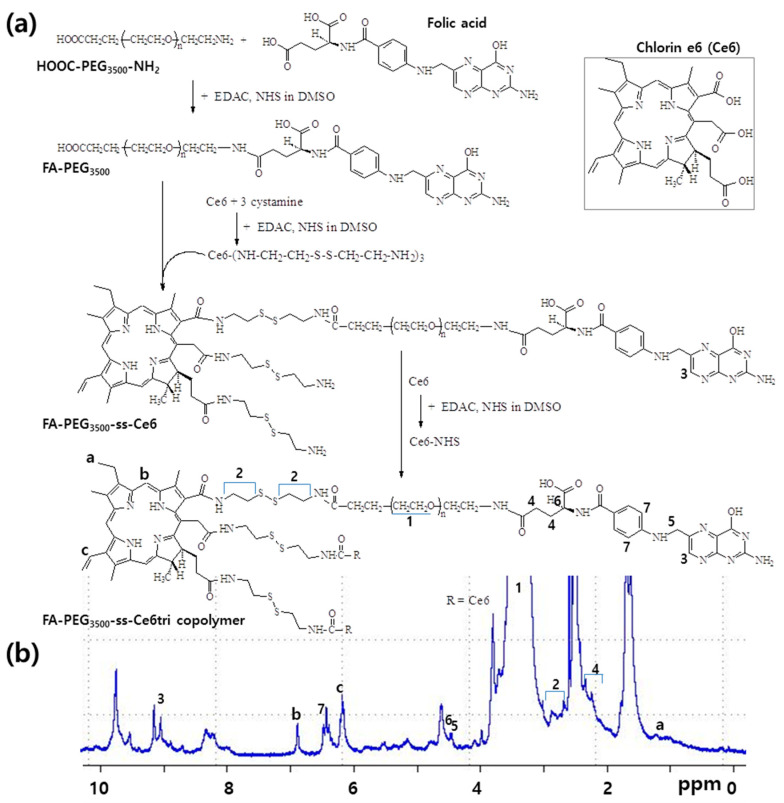
(**a**) Synthesis scheme and (**b**) ^1^H NMR spectra of FA–PEG_3500_-ss-Ce6tri copolymer. FA–PEG_3500_-ss-Ce6tri copolymer in dimethyl sulfoxide (DMSO)-d form was measured with ^1^H NMR spectroscopy.

**Figure 2 materials-13-02810-f002:**
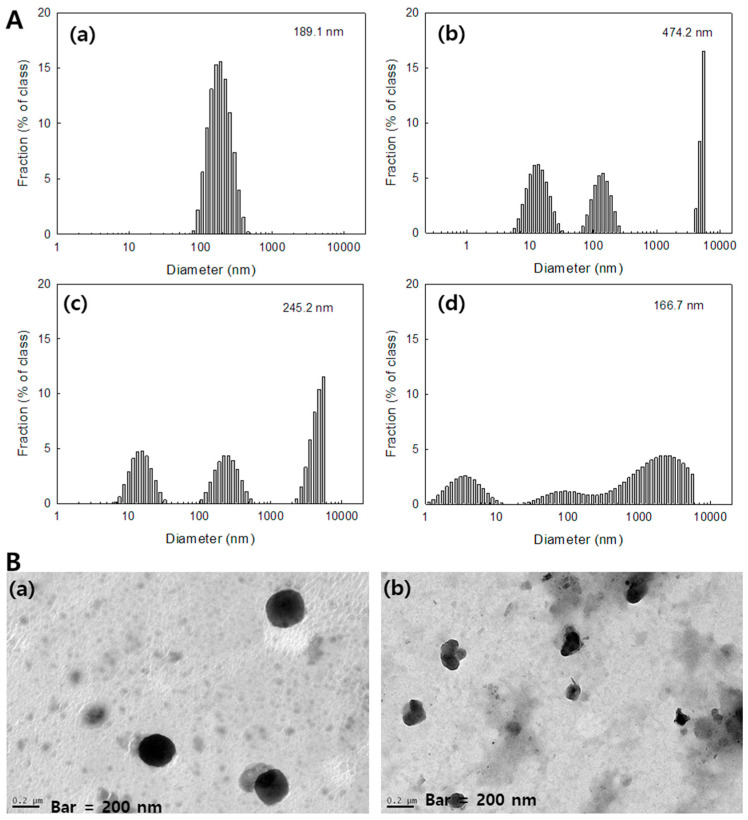
(**A**) Particle-size distribution of FA–PEG_3500_-ss-Ce6tri copolymer nanophotosensitizers. GSH concentration: (**a**) L-glutathione reduced (GSH), 0 mM; (**b**) GSH, 5 mM; (**c**) GSH, 10 mM; (**d**) GSH, 20 mM; (**B**) morphologic observations of FA–PEG_3500_-ss-Ce6tri copolymer nanophotosensitizers. (**a**) GSH, 0 mM; (**b**) GSH, 10 mM.

**Figure 3 materials-13-02810-f003:**
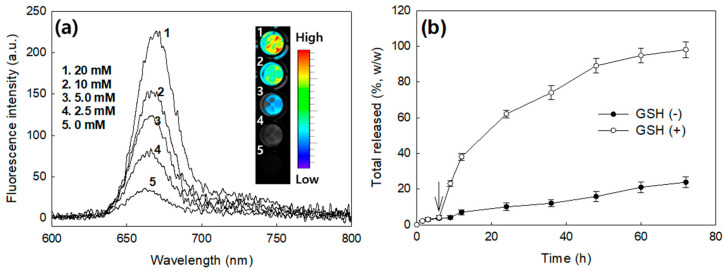
(**a**) Effect of GSH concentration on the fluorescence spectra of FA–PEG_3500_-ss-Ce6tri copolymer nanophotosensitizers. For fluorescence images, nanophotosensitizer solution (0.1 mg/mL) in PBS (pH 7.4, 0.01 M) was incubated at 37 °C for 3 h with GSH; (**b**) Ce6 release from FA–PEG_3500_-ss-Ce6tri copolymer nanophotosensitizers in the presence or absence of GSH. GSH concentration in the media was 0 mM for GSH(-) and 20 mM for GSH(+).

**Figure 4 materials-13-02810-f004:**
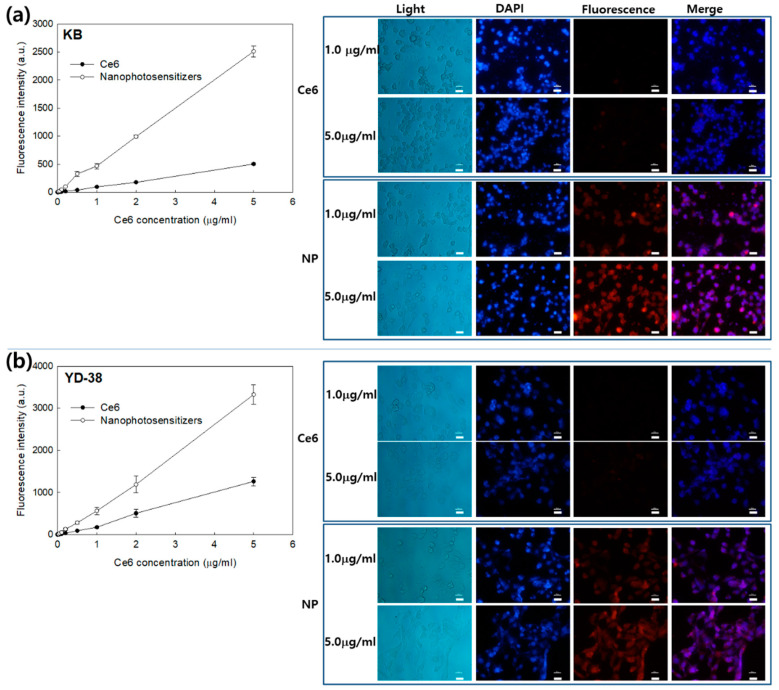
Ce6 uptake ratio (left panel) and fluorescence microscopic observation (right panel) of Ce6 alone and FA–PEG_3500_-ss-Ce6tri copolymer nanophotosensitizers. (**a**) KB cells; (**b**) YD-38 cells. 2 × 10^4^ cells were treated with various concentrations of Ce6 or nanophotosensitizers for 2 h. Bar = 20 μm.

**Figure 5 materials-13-02810-f005:**
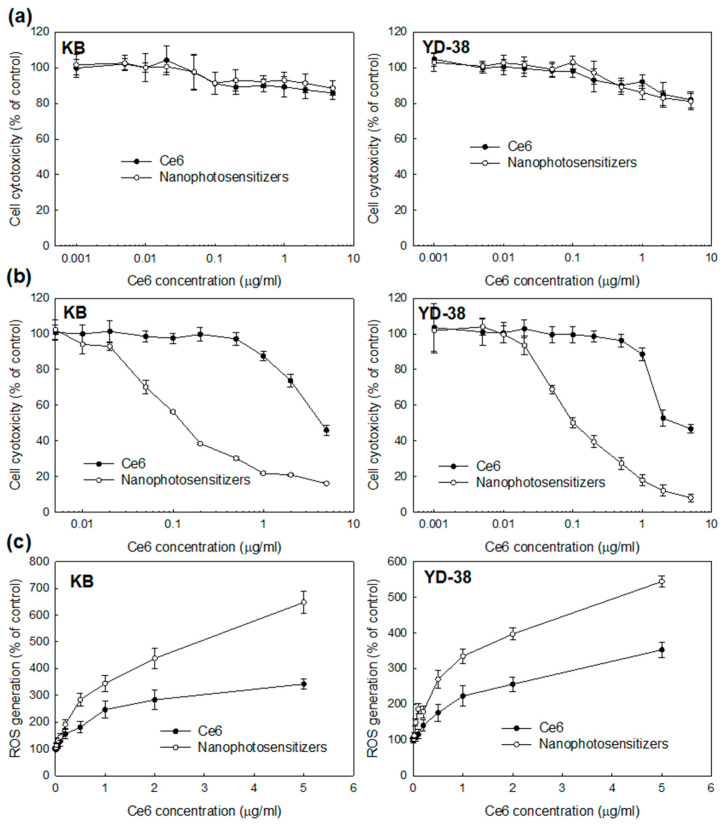
Photodynamic therapy (PDT) efficacy of FA–PEG_3500_-ss-Ce6tri copolymer nanophotosensitizers. (**a**) Dark toxicity; (**b**) Phototoxicity; (**c**) reactive oxygen species (ROS) production. Dark toxicity and phototoxicity of KB cells and YD-38 cells were evaluated with 3-(4,5-dimethylthiazol-2-yl)-2,5-diphenyltetrazolium bromide (MTT) proliferation assay. For phototoxicity, cells were treated with Ce6 or nanophotosensitizers for 2 h and, after that, irradiated at 664 nm for 9 min 29 s (2.0 J/cm^2^). For dark toxicity, cells were treated with Ce6 or nanophotosensitizers similarly in the absence of irradiation.

**Figure 6 materials-13-02810-f006:**
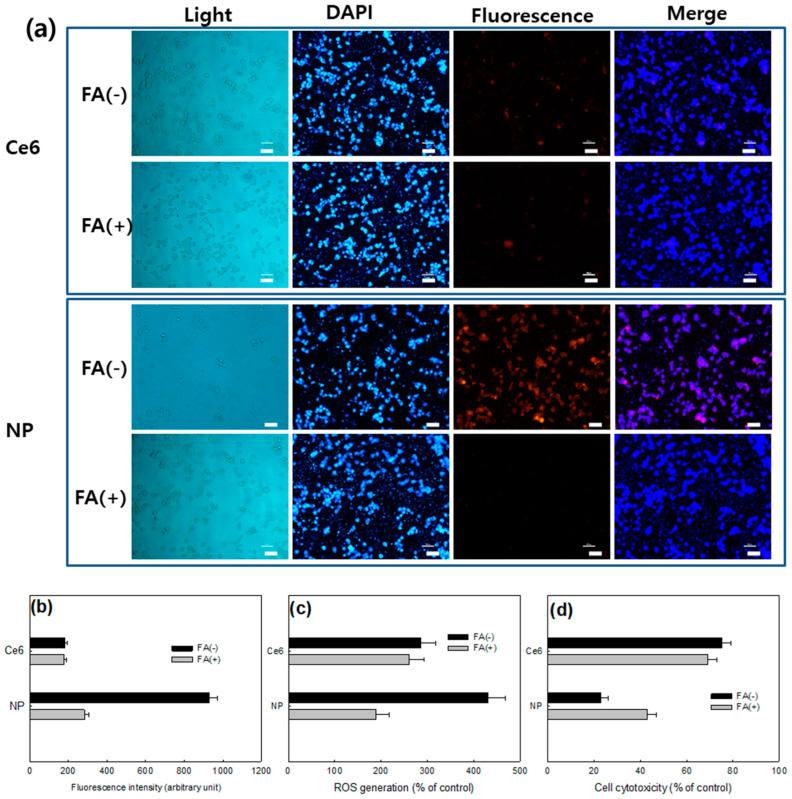
(**a**) Fluorescence microscopic observations of KB cells treated with Ce6 alone or FA–PEG_3500_-ss-Ce6tri copolymer nanophotosensitizers. FA was pretreated to the cells to block folate receptor; (**b**) Ce6 uptake ratio; (**c**) ROS production; (**d**) phototoxicity. Competition assay for folate receptor-sensitivity of KB cells, cells were exposed to 5-mM folic acid for 1 h and, after that, cells were washed Ce6 alone or nanophotosensitizers (2 μg/mL as a Ce6) were treated to cells. Following procedures was similar to those of Figure 4 and Figure 5. Bar = 40 μm.

**Figure 7 materials-13-02810-f007:**
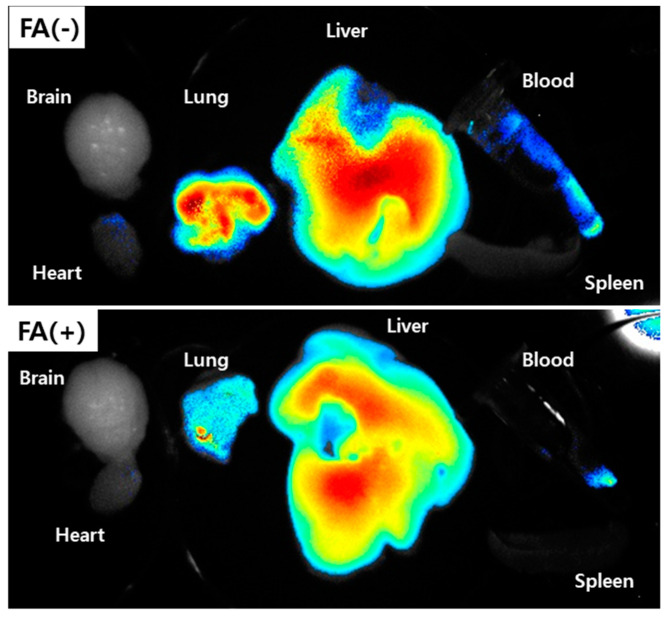
Pulmonary metastasis of KB cells. To block folate receptor of KB cells, folic acid in PBS (dose: 20 mg/kg) was i.v. administered 1 h before injection of nanophotosensitizer. 1 × 10^6^ KB cells were intravenously i.v. administered via the tail vein of nude BALb/C mouse. Three weeks later, nanophotosensitizers (dose: 10 mg/kg) were i.v. administered via the tail vein of the mouse. To block folate receptor (FA(+)), folic acid in PBS (dose: 20 mg/kg) was i.v. administered 1 h before nanophotosensitizer administration. PBS without FA was i.v. administered for comparison (FA(-)). To observe fluorescence images of each organ, the mice were sacrificed 24 h later.

**Table 1 materials-13-02810-t001:** Characterization of FA–PEG_3500_ and Ce6-conjugated copolymer.

	Ce6 Contents (%, *w/w*) ^a^		Particle Size (nm) ^b^
	Theoretical	Experimental	
FA–PEG_3500_ conjugates	-	-	-
FA–PEG_3500_-ss-Ce6	12.2	11.8	96.2 ± 7.3
FA–PEG-ss-Ce6tri	29.5	28.3	189.1 ± 10.8

^a^ Ce6 contents (%, *w/w*) = w of Ce6/w of copolymer. Ce6 contents in the copolymer were evaluated as described in 2.6. Ce6 release study: 5 mg of copolymer in 50 mL PBS was incubated with 1,4-dithiothreitol (DTT) for 24 h and then diluted with DMSO ten times. ^b^ Particle size: average ± standard deviation (S.D.) from three measurements.

**Table 2 materials-13-02810-t002:** IC_50_ values of Ce6 and nanophotosensitizers.

	IC_50_ (μg/mL) ^a^	
	KB Cells	YD-38 Cells
Ce6	4.52	3.13
Nanophotosensitizers	0.13	0.12

^a^ IC_50_ values of each cells were evaluated from phototoxicity in Figure 5b.

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
