# Peer review of "Nanophotosensitizers for Folate Receptor-Targeted and Redox-Sensitive Delivery of Chlorin E6 against Cancer Cells"

_materials, 2020, doi:10.3390/ma13122810_

Round 1

Reviewer 1 Report

This manuscript reports the development of new nanoparticles based on chlorin e6-PEG-folate conjugates for selective targeting of folate receptor-tumor cells and redox responsive release of photosensitizer Ce6 to enhance the PDT efficiency. This novel nanophotosensitizer (FA-PEG-ss-Ce6tri) has been designed and studied in oral cancer cell lines, KB human epithelial carcinoma and YD-38 squamous carcinoma cells. The in vitro studies with this conjugate demonstrated a significant increase in cellular uptake, ROS production and phototoxicity when compared to Ce6 alone. Also, preliminary in vivo results confirmed that FA-PEG-ss-Ce6tri has the ability to target the folate receptor. 

This work suggests that FA-PEG-ss-Ce6tri is a good platform for selective delivery of photosensitizer Ce6 into tumors.

The following issues need addressing:

The authors should carefully revise the text and the writing improved. Articles (like "the", "a", "an") are often missing and several times the plural and singular forms are being used incorrectly. Examples:  Line 30 - Replace “candidate” with “candidates”; Line 52 - Replace “these” with “this”; Line 53 - Replace “was” with “were”; Line 136 - Replace “in” with “at”; Line 167 - Replace “was” with “were”; Line 245 - “Replace was” with “were”; Line 275 - Replace “Figure” with “Figures”

Throughout the manuscript, the authors should use a space between a value and the units used. Examples: Lines 90 (20 mg), 93 (5 L), 146 (37 °C), 150 (1 mg/ml). Furthermore, it would be clearer to specify the Polyethylene Glycol used, e.g. PEG3500

Line 2 - Receptor targeted is missing a hyphen “Receptor-targeted”.

Lines 26-28 - In the abstract, the authors state that the copolymer nanophotosensitizers present higher activity (uptake, ROS, cytotoxicity).  Compared to what?

Lines 45-48 – The statement: “…PDT  treatment  for  cancer  is  able  to  minimize  adverse  side   effects  because photosensitizers produces excess amount of reactive oxygen species (ROS) under the irradiation condition with specific wavelength of light and they have negligible cytotoxicity against surrounding normal tissues and cells, i.e. diseased cells can be eliminated only irradiated field”. The use of the expression “excess amount of ROS” seems to indicate that the production of ROS was excessive. What do the authors mean by “diseased cells can be eliminated only irradiated field”? This sentence should be rephrased in a clearer way.

Lines 57-59 – I suggest replace “it can be provoked phototoxicity problem, which it requires blocking of sun light” with “ it causes sunlight sensitivity, and its exposure should be avoided.

Line 114 - Replace “was defined with” with “were elucidated by”.

Line 136 - Add a hyphen to “film-coated”.

Lines 220-243 - To make the synthesis more reader-friendly I suggest that the main chemical structures in Figure 1 should be labelled with their abbreviations and used throughout the text. Figure 1 contains some chemical errors as well. The reaction of the carboxylic acid group of FA with the NH2-terminal residue of PEG forms an amide. Also, the chemical structure of chlorin e6 is incorrect, NH protons and double bonds are missing.

Line 261 – Replace “redox sensitive” with “a redox-sensitive”.

Line 279 – Replace “miscopy” with “microscopy”.

Author Response

Answer to reviewer 1’s comment

This manuscript reports the development of new nanoparticles based on chlorin e6-PEG-folate conjugates for selective targeting of folate receptor-tumor cells and redox responsive release of photosensitizer Ce6 to enhance the PDT efficiency. This novel nanophotosensitizer (FA-PEG-ss-Ce6tri) has been designed and studied in oral cancer cell lines, KB human epithelial carcinoma and YD-38 squamous carcinoma cells. The in vitro studies with this conjugate demonstrated a significant increase in cellular uptake, ROS production and phototoxicity when compared to Ce6 alone. Also, preliminary in vivo results confirmed that FA-PEG-ss-Ce6tri has the ability to target the folate receptor. 

This work suggests that FA-PEG-ss-Ce6tri is a good platform for selective delivery of photosensitizer Ce6 into tumors.

The following issues need addressing:

The authors should carefully revise the text and the writing improved. Articles (like "the", "a", "an") are often missing and several times the plural and singular forms are being used incorrectly. Examples:  Line 30 - Replace “candidate” with “candidates”; Line 52 - Replace “these” with “this”; Line 53 - Replace “was” with “were”; Line 136 - Replace “in” with “at”; Line 167 - Replace “was” with “were”; Line 245 - “Replace was” with “were”; Line 275 - Replace “Figure” with “Figures”

Answer) Thanks for your valuable comment. According to your comment, we revised the manuscript and grammatical problems were also solved by aid of English-editing service.

Throughout the manuscript, the authors should use a space between a value and the units used. Examples: Lines 90 (20 mg), 93 (5 L), 146 (37 °C), 150 (1 mg/ml). Furthermore, it would be clearer to specify the Polyethylene Glycol used, e.g. PEG3500

Answer) Thanks for your valuable comment. According to your comment, we corrected the units and PEG3500.

Line 2 - Receptor targeted is missing a hyphen “Receptor-targeted”.

Answer) Thanks for your valuable comment. According to your comment, we corrected and added hyphen.

Nanophotosensitizers for folate receptor-targeted and redox-sensitive delivery of chlorin e6 against cancer cells

Lines 26-28 - In the abstract, the authors state that the copolymer nanophotosensitizers present higher activity (uptake, ROS, cytotoxicity).  Compared to what?

Answer) Thanks for your valuable comment. These are corrected to following phrases:

Compared to Ce6 alone, FA-PEG3500-ss-Ce6tri copolymer nanophotosensitizers recorded higher Ce6 uptake ratio, ROS production and cellular cytotoxicity against KB and YD-38 cells.

Lines 45-48 – The statement: “…PDT  treatment for cancer is able to minimize adverse side effects because photosensitizers produces excess amount of reactive oxygen species (ROS) under the irradiation condition with specific wavelength of light and they have negligible cytotoxicity against surrounding normal tissues and cells, i.e. diseased cells can be eliminated only irradiated field”. The use of the expression “excess amount of ROS” seems to indicate that the production of ROS was excessive. What do the authors mean by “diseased cells can be eliminated only irradiated field”? This sentence should be rephrased in a clearer way.

Answer) Thanks for your valuable comment. Practically, ROS can be generated where specific wavelength of light was irradiated. This means that non-irradiated field does not produces ROS by Ce6

Since photosensitizers produces ROS only if specific wavelength of light is irradiated, PDT can be specifically affected to the tumor tissues with minimized undesirable side effects against surrounding normal tissues and cells [8-10].

Lines 57-59 – I suggest replace “it can be provoked phototoxicity problem, which it requires blocking of sun light” with “ it causes sunlight sensitivity, and its exposure should be avoided.

Answer) Thanks for your valuable comment. According to your comment, those phrases were corrected to “Even though photosensitizer itself is activated under the specific wavelength of light, it causes sunlight sensitivity, and its exposure should be avoided [22].”

Line 114 - Replace “was defined with” with “were elucidated by”.

Answer) Thanks for your valuable comment. According to your comment, we corrected those phrases to “The structure of synthesized conjugates were elucidated by”

Line 136 - Add a hyphen to “film-coated”.

Answer) Thanks for your valuable comment. According to your comment, we corrected manuscript and add hyphen.

Lines 220-243 - To make the synthesis more reader-friendly I suggest that the main chemical structures in Figure 1 should be labelled with their abbreviations and used throughout the text. Figure 1 contains some chemical errors as well. The reaction of the carboxylic acid group of FA with the NH2-terminal residue of PEG forms an amide. Also, the chemical structure of chlorin e6 is incorrect, NH protons and double bonds are missing.

Answer) Thanks for your valuable comment. According to your comment, we corrected Figure 1 and added labels.

Line 261 – Replace “redox sensitive” with “a redox-sensitive”.

Answer) Thanks for your valuable comment. According to your comment, we corrected manuscript and add hyphen.

Line 279 – Replace “miscopy” with “microscopy”.

Answer) Thanks for your valuable comment. According to your comment, we corrected this word in the manuscript.

Reviewer 2 Report

The paper “Nanophotosensitizers for folate receptor targeted and redox-sensitive delivery of chlorin e6 against cancer cells” by Min-Suk Kook and others, presents an interesting approach to drug delivery of Ce6 in the context of PDT, that is the use of copolymer nanophotosensitizers.  

The topic is interesting for drug delivery research and pre-clinical research on PDT. The controlled release and penetration in tumors tissues is a particular challenge, and the authors are commended for their attempt to address this problem with their copolymer nanophotosensitizers. However, the manuscript needs extensive improvements, mainly in the introduction and discussion section. Otherwise the study appears technically sound and appropriate for publication when the following issues are addressed.

Main points:

  1. Please improve the use of English language throughout the manuscript. This should be done by a native speaker or someone with a high degree of professional fluency. On various occasions it was difficult/impossible to interpret the text and decipher the meaning of certain passages.
  2. The introduction fails to capture the relevance and impact of this study. This study deals with Ce6 vectorization and its targeting with FA against oral cancer cells. Therefore, we need literature background concerning Ce6 vectorization (there is no need to mention photofrin or 5-Ala). Similarly, only 2 lines deal with folic acid at the very end of the introduction. Up to line 60 you only describe the generalities of PDT, this introductory part is too developed to the detriment of the other parts. 
  3. Similarly, the discussion lacks relevance. From line 328 to 351, you repeat in too much detail the generalities already mentioned in the introduction. We must wait for line 352 to start finding a discussion of your results.
  4. The novelty of the use of FA-PEG-ss-Ce6tri needs to be better described. Have such delivery systems been used and/or developed before?

Minor points:

  1. Fig 5b : it would be appropriate to calculate IC50s in order to be able to compare the survival of different cell lines and different molecules.
  2. More information should be provided on the cell lines, including their folic acid receptor status.
  3. Scale bars on microscopy images are not readable.
  4. Prefer a Y-axis scale such as “ROS generation” rather than “Fluorescence intensity” (Fig 5c; Fig 6c)
  5. Specify in the legend of figure 7 if these are representative images and how many animals were used for this experiment.

Author Response

Answer to reviewer 2’s comment

The paper “Nanophotosensitizers for folate receptor targeted and redox-sensitive delivery of chlorin e6 against cancer cells” by Min-Suk Kook and others, presents an interesting approach to drug delivery of Ce6 in the context of PDT, that is the use of copolymer nanophotosensitizers.  

The topic is interesting for drug delivery research and pre-clinical research on PDT. The controlled release and penetration in tumors tissues is a particular challenge, and the authors are commended for their attempt to address this problem with their copolymer nanophotosensitizers. However, the manuscript needs extensive improvements, mainly in the introduction and discussion section. Otherwise the study appears technically sound and appropriate for publication when the following issues are addressed.

Main points:

  1. Please improve the use of English language throughout the manuscript. This should be done by a native speaker or someone with a high degree of professional fluency. On various occasions it was difficult/impossible to interpret the text and decipher the meaning of certain passages.

Answer) Thanks for your valuable comment. According to your comment, we revised the manuscript and English expression was also improved by aid of native speaker.

  1. The introduction fails to capture the relevance and impact of this study. This study deals with Ce6 vectorization and its targeting with FA against oral cancer cells. Therefore, we need literature background concerning Ce6 vectorization (there is no need to mention photofrin or 5-Ala). Similarly, only 2 lines deal with folic acid at the very end of the introduction. Up to line 60 you only describe the generalities of PDT, this introductory part is too developed to the detriment of the other parts. 

Answer) Thanks for your valuable comment. According to your comment, we fully revised the manuscript in introduction part and discussion part. We add introduction for the PDT for oral cancer and the use of folic acid for drug targeting issues. Furthermore, Discussion part also fully revised according to your comment.

  1. Introduction

The incidence rate and mortality of oral cancer has been continuously increasing worldwide in the last two decades. Oral cancer patients are known to have the lowest survival rate marked at five-year among major cancer types.[1]. Oral cancers are frequently diagnosed in their advanced stages and when they are metastasized to other regions despite its easy accessibility for identification of a cancer lesion [2,3]. Even though surgery, chemotherapy, radiotherapy, and immunotherapy are currently considered as the treatment for the oral cancer, high recurrence rate and metastasis to the distant regions after sole or combined treatment are still problematic and resulted in low survival rates [4-7]. Therefore a novel treatment regimen for oral cancers is required to improve therapeutic efficacy with reduced side effects.

Since photodynamic therapy (PDT) is composed of safe components such as light, oxygen species, and photosensitizer, it has been believed to be a promising candidate for cancer treatment [8,9]. PDT treatment for cancer can minimize adverse side effects because photosensitizers produce an excess amount of reactive oxygen species (ROS) under the irradiation condition of specific wavelength of light and they have negligible cytotoxicity against surrounding normal tissues and cells, which means cells only in irradiated field can be eliminated [8-10]. Many researchers and clinicians have been trying to eradicate malignant or premalignant tumors using PDT [9-15]. It has been speculated that PDT is a suitable treatment option for oral cancers because oral cancers including gingival cancer occur in oral cavity, of which can be easily accessed via visible light for PDT treatment, and, from these point of view, it has been used in control of early or advanced stages of oral cancers [15-19]. Nevertheless, Nauta et al., reported that unsatisfactory long-term results were observed in PDT on squamous cell carcinoma using porfimer sodium and PDT efficacy was dependant on the wavelength of light [17]. Furthermore, it is known that the drug-resistant problem in oral cancer cells also occurs in PDT with 5-aminolevulinic acid (5-ALA) [20]. Traditional photosensitizers such as 5-ALA also have limited tumor specificity followed by the problem of them spreading throughout the body [10, 22]. This is because, even though photosensitizer itself is activated under the specific wavelength of light, phototoxicity can be provoked, so it is required to block the sun light [22]. Photosensitizers should be concentrated in the diseased tissues or specific cells to minimize these problems.

Polymeric- or nano-photosensitizers have been investigated to solve problems of traditional photosensitizers since nanoparticles have can be used to deliver the anticancer drug to the specific site of action because nanocarriers can be used to deliver the anticancer drug to the specific site of action [23-28]. Nanocarriers emphasize anticancer activity of traditional anticancer drugs and can be used for diagnostic/therapeutic purposes [23,24]. Especially, nanoparticles decorated with targeting molecules on the surface have great potential to target cancer cells because cancer cells express many kinds of molecular receptors excessively such as CD44 and folate receptors [25-28]. For example, hyaluronic acid (HA) can recognize CD44 receptors on the cancer cells and then they can be used to detect CD44-overexpressing cancer cells through conjugation with Ce6 [25,26]. Lee and Jeong reported that HA-Ce6-poly(L-histidine) copolymer nanoparticles are able to deliver the anticancer drugs through CD44-mediated pathway against CD44-overexpressing cancer cells and then efficiently inhibited growth of tumor [23]. Furthermore, folate receptors on the cell surface also can be applied to target cancer cells since expression of folate receptors is known to be increased according to the malignancies of tumor [27,28]. Folic acid (FA)-conjugated polymer nanoparticles concentrated more photosensitizers or anticancer drugs into tumor tissue through the folate-receptor-mediated pathway compared to the normal tissue [29,30]. Folate receptor-mediated drug targeting using nanoparticles may solve drawbacks of traditional photosensitizers.

  1. Similarly, the discussion lacks relevance. From line 328 to 351, you repeat in too much detail the generalities already mentioned in the introduction. We must wait for line 352 to start finding a discussion of your results.

Answer) Thanks for your valuable comment. According to your comment, we fully revised the manuscript in introduction part and discussion part. In discussion part, we discussed more about our study and compared with other studies. And, we also cited more other references.

  1. Discussion

Furthermore, oral cancer such as gingival cancers is difficult to cure because traditional regimens for treatment have limited efficacy and difficulties for diagnosis [2,3]. Even though surgery, chemotherapy, radiotherapy, and immunotherapy are currently considered as the treatment for the oral cancer, high recurrence rate and metastasis to the distant regions after sole or combined treatment are still problematic and resulted in low survival rates [4-7]. From this point of view, PDT regimens can be considered as a promising candidate. However, drawbacks of PDT such as sun-shade problem limit its clinical application [22]. If photosensitizers can be preferentially delivered to tumors rather than surrounding normal tissues, its fluorescence properties can be used to detect and treat tumors specifically [8]. Actually, we showed that nanophotoseensizers can be specifically delivered to tumor cells via folate receptor-mediated manner as shown in Figure 6 and 7. That is, FA-PEG3500-ss-Ce6tri nanophotosensitizers were delivered preferentially through folate receptor specific manner against KB cells as shown in Figure 6. In vivo pulmonary metastasis model of KB cells also proved that FA-PEG3500-ss-Ce6tri nanophotosensitizers can be delivered and concentrated in the tumor cells by folate receptor specific manner. These results mean that photosensitizers can be specifically delivered to the cancer cells via specific receptors of oral cancer cells and then concentrated in the tumor tissues rather than normal cells. Li et al. also reported that photosensitizer conjugated with FA-PEG can be delivered by folate receptor-specific manner against folate-receptor-overexpressing cancer cells [31]. Especially, folate receptor expression is known to be increased in the patients having periodontal disease [32]. Alkan et al., reported that expression values of folate-receptor 1 was higher in gingivitis and periodontitis groups than those of healthy group [32].

Meanwhile, reduction-oxidation (redox)-potential is also known to be elevated in the patients having periodontal disease [33,34]. Wong et al., reported that GSH level in oral cancer tissues was significantly higher than that of adjacent non-tumor tissue parts and its level has co-relationship with tumor size [34]. We showed that FA-PEG3500-ss-Ce6tri nanophotosensitizers were specifically disintegrated by GSH-dependent manner and then Ce6 can be liberated from nanophotosensitizers by response to the concentration of GSH as shown in Figure 3. Since intracellular level of GSH in cancer cells is higher than extracellular level, GSH-dependent liberation of Ce6 may be considered for intracellular targeting of photosensitizers [35,36]. The fact that tumor cells have higher redox-potential that adjacent normal cells and tissues can be also used to detect tumor [37,38]. PDT may offer new approaches for early detection and treatment of oral cancers. Mallia et al. reported that ALA-induced normalized fluorescence was applied to detect anatomical locations of the oral cavity to improve the diagnostic contrast and accuracy of oral cancers [37]. In addition to that, PDT can be used to non-invasive treatment of precancerous lesions. Maloth et al. reported that positive response was obtained from PDT approach against oral leukoplakia lesions oral lichen planus lesions [30]. Since traditional photosensitizers have low tumor-specificity and also disperse to the normal cells or tissues, however, tumor-specific delivery of photosensitizers may offer improved diagnosis and treatment regimen of oral cancers. FA-PEG3500-ss-Ce6tri nanophotosensitizers have redox-sensitive and folate-receptor sensitive behavior against oral cancer cells as shown in Figure 3, 6, and 7. These results may enable us to detect and treat oral cancer cells with minimal side-effects against normal cells/tissues. Especially, tumor tissue-specific cytotoxicity during PDT can be triggered by irradiation of specific disease sites even though photosensitizers are fully distributed around the human body because PDT with did not reveal cellular cytotoxicity in the absence of light irradiation [39].

In conclusion, we synthesized FA-PEG3500-ss-Ce6tri copolymer to deliver photosensitizers against cancers with redox-sensitive and folate-specific manner. FA-PEG3500-ss-Ce6tri copolymer nanophotosensitizers have small particle sizes of less than 200 nm and spherical shapes. They were disintegrated and Ce6 released by GSH-dependent manner. Higher Ce6 uptake ratio, ROS production and cellular cytotoxicity against KB and YD-38 cells were obtained by treatment of FA-PEG3500-ss-Ce6tri copolymer nanophotosensitizers. Furthermore, nanophotosensitizers were delivered by folate receptor-specific manner in vitro and in vivo. We suggest that FA-PEG3500-ss-Ce6tri copolymer nanophotosensitizers are the promising candidate for PDT of oral cancers.

Sun, X.; Sun, J.; Lv, J.; Dong, B.; Liu, M.; Liu, J.; Sun, L.; Zhang, G.; Zhang, L., Huang, G.; Xu, W.; Xu, L.; Bai, X.; Song, H. Ce6-C6-TPZ co-loaded albumin nanoparticles for synergistic combined PDT-chemotherapy of cancer. J. Mater. Chem. B. 2019, 7, 5797-5807.

Wang, B.Y.; Liao, M.L.; Hong, G.C.; Chang, W.W.; Chu, C.C. Near-Infrared-Triggered Photodynamic Therapy toward Breast Cancer Cells Using Dendrimer-Functionalized Upconversion Nanoparticles. Nanomaterials 2017, 7, 269.

Li, D.; Li, P.; Lin H.; Jiang, Z.; Guo, L.; Li, B. A novel chlorin–PEG–folate conjugate with higher water solubility, lower cytotoxicity, better tumor targeting and photodynamic activity. J. Photochem. Photobiol. B. 2013, 127, 28-37.

Alkan, D.; Guven, B.; Turer, C.C.; Balli, U.; Can, M. Folate-receptor 1 level in periodontal disease: a pilot study. BMC Oral Health. 2019, 19, 218.

Kenney, E.B.; Ash Jr, M.M. Oxidation Reduction Potential of Developing Plaque, Periodontal Pockets and Gingival Sulci. J Periodontol. 1969, 40, 630-633.

Wong, D.Y.; Hsiao, Y.L.; Poon, C.K.; Kwan, P.C.; Chao, S.Y.; Chou, S.T.; Yang, C.S. Glutathione concentration in oral cancer tissues. Cancer Let. 1994, 81, 111-116.

Park, H.; Na, K. Conjugation of the Photosensitizer Chlorin e6 to Pluronic F127 for Enhanced Cellular Internalization for Photodynamic Therapy. Biomaterials. 2013, 34, 6992-7000.

  1. The novelty of the use of FA-PEG-ss-Ce6tri needs to be better described. Have such delivery systems been used and/or developed before?

Answer) Thanks for your valuable comment. Actually, FA-PEG conjugates having photosensitizers were already investigated by some of researchers. For example, Li et al., reported that FA-PEG was conjugated with chlorin molecules (Not Ce6) for improvement of aqueous solubility and cancer cell delivery efficacy. They also obtained positive results from these conjugates using folate-receptor-overexpressing cells. However, they simply conjugated with one chlorin molecule with FA-PEG. Then, chlorin contents in one polymer molecule (FA-PEG-chlorin) were lower than that of our conjugates because our conjugates have Ce6 trimer in on polymer molecule. Furthermore, their conjugates have no disulfide linkages (for redox sensitive delivery of Ce6). In our case, we used Ce6 trimer to synthesize polymer conjugates. Then we would like to claim that FA-PEG3500-Ce6tri copolymer conjugates have superior potential in PDT approaches that those of other reports. Anyway, we discussed more for this issues and added references.

The fact that tumor cells have higher redox-potential that adjacent normal cells and tissues can be also used to detect tumor [38,39]. Even though FA-PEG-photosensitizer conjugates reported by Li et al. also have a targeting potential against cancer cells, their conjugates have no redox-sensitive linkages and then simply provide folate-receptor sensitivity [32]. In our system, Ce6 domain in the FA-PEG3500-ss-Ce6tri nanophotosensitizers is existed as a timer, i.e. Ce6 molecules are linked with each other via disulfide linkages. These properties of FA-PEG3500-ss-Ce6tri copolymer may provide superior redox-sensitivity of nanophotosensitizers as shown in Figure 3.

  1. Stallivieri, A.; Colombeau, L.; Jetpisbayeva, G.; Moussaron, A.; Myrzakhmetov, B.; Arnoux, P.; Acherar, S.; Vanderesse, R.; Frochot, C. Folic acid conjugates with photosensitizers for cancer targeting in photodynamic therapy: Synthesis and photophysical properties. Bioorg. Med. Chem. 2017, 25, 1-10.
  2. Li, D.; Li, P.; Lin H.; Jiang, Z.; Guo, L.; Li, B. A novel chlorin–PEG–folate conjugate with higher water solubility, lower cytotoxicity, better tumor targeting and photodynamic activity. J. Photochem. Photobiol. B. 2013, 127, 28-37.

Minor points:

  1. Fig 5b : it would be appropriate to calculate IC50s in order to be able to compare the survival of different cell lines and different molecules.

Answer) Thanks for your valuable comment. According to your comment, we evaluated IC50s of Figure 5b and summarized in Table 2.

As shown in Table 2, IC50 values of nanophotosensitizers were significantly lower than that of Ce6, indicating that nanophotosensitizers have superior PDT efficacy compared to Ce6 alone.

Table 2. IC50 values of Ce6 and nanophotosensitizers.

IC50 (mg/ml)a

KB cells

YD-38 cells

Ce6

Nanophotosensitizers

4.52

0.13

3.13

0.12

a IC50 values of each cells were evaluated from phototoxicity in Figure 5(b).

  1. More information should be provided on the cell lines, including their folic acid receptor status.

Answer) Thanks for your valuable comment. Many references reported that KB cells are expressed folate receptor and then many scientists use folate receptor for drug targeting issues. However, there are no information whether or not YD-38 cells are expressed folate receptor. Then we used KB cells for folate receptor sensitivity. Anyway, we added the reference.

Weitman, S.D.; Lark, R.H.; Coney, L.R.; Fort, D.W.; Frasca, V.; Zurawski Jr, V.R.; Kamen, B.A.; Distribution of the Folate Receptor GP38 in Normal and Malignant Cell Lines and Tissues. Cancer Res, 1992, 52, 3396-3401.

  1. Scale bars on microscopy images are not readable.

Answer) Thanks for your valuable comment. According to your comment, we added magnification status in the Figure legend. Practically, we observed the cells at 200x or 400x and then added magnification status in the Figure legend.

  1. Prefer a Y-axis scale such as “ROS generation” rather than “Fluorescence intensity” (Fig 5c; Fig 6c). Specify in the legend of figure 7 if these are representative images and how many animals were used for this experiment.

Answer) Thanks for your valuable comment. According to your comment, we change the Y-axis legend (Fluorescence intensity) to ROS generation in Figure 5 and 6. Practically, we used 2 animals for each group and we confirmed similar tendency from biodistribution study. Then, we present images one of them.

Figure 5                       Figure 6

Reviewer 3 Report

Dear editor,

This paper describes the polymer-based nanoparticles covalently connected with folate-targeting ligands and photosensitizers (Ce6) toward cancer therapy. The Ce6 can be released from the nanoparticles upon GSH-induced reduction, and two cancer cell lines were effectively killed upon light excitation. This hybrid system shows a promising character as the nanophotosensitizer, I can recommend this paper to be published in the journal of “Materials” after some problems are revised.

  1. The quality of Figure 1 should be improved. The chemical structures in the synthesis scheme must be clear, and resolution for the NMR spectrum should be improved. The authors should assign all the relating protons in the copolymer structures in the NMR spectrum. Molecular weights (MS) analysis is necessary to provide more solid evidence for the drawn chemical structures. If the final material is a mixture, the authors should discuss it in the text.
  2. How to identify the Ce6 contents in Table 1? The authors did not provide the information in the main text.
  3. In Figure 2, the copolymer was treated with GSH to form a complex structure. The size distribution becomes larger and wider, but the TEM image only shows a collapsed structure and the size became smaller. The authors should explain this point.
  4. In Figure 3, the authors should address the reason regarding the fluorescence increment of the Ce6 released from the nanoparticles upon GSH treatment.
  5. The PDT efficacy of Ce6 alone and nanophotosensitizers at different concentrations were compared in Figure 5. How to determine the Ce6 concentration loaded onto the nanoparticles?
  6. Recent articles regarding the PDT therapy of Ce6-loaded hybrid nanoparticles can be cited: for example, Wang, BY et al. Nanomaterials 2017, 7, p269; Sun X et al. J. Mater. Chem. B, 2019, 38, p5797.

Author Response

Answer to reviewer 3’s comment

This paper describes the polymer-based nanoparticles covalently connected with folate-targeting ligands and photosensitizers (Ce6) toward cancer therapy. The Ce6 can be released from the nanoparticles upon GSH-induced reduction, and two cancer cell lines were effectively killed upon light excitation. This hybrid system shows a promising character as the nanophotosensitizer, I can recommend this paper to be published in the journal of “Materials” after some problems are revised.

  1. The quality of Figure 1 should be improved. The chemical structures in the synthesis scheme must be clear, and resolution for the NMR spectrum should be improved. The authors should assign all the relating protons in the copolymer structures in the NMR spectrum. Molecular weights (MS) analysis is necessary to provide more solid evidence for the drawn chemical structures. If the final material is a mixture, the authors should discuss it in the text.

Answer) Thanks for your valuable comment. We revised Figure 1 according to your comment. Molecular structures were assigned and indicated with NMR spectra in Figure 1. At this moment, we are going to analyze molecular weights of FA-PEG3500-ssCe6tri copolymer nanophotosensitizers with various equipment and we will report it in the next paper with other characterization results. Anyway, when we calculate Ce6 contents in the copolymer was almost similar to those of theoretical values. These results indicated that Ce6 trimer must be formed in the copolymer structure. Practically, unbound Ce6 was removed from the copolymer solution by dialysis procedure. We assumed that Ce6 trimer formed in the copolymer structure.

Experimental contents of Ce6 were almost similar to the theoretical value in the copolymer, indicating that Ce6 trimer must be formed in the copolymer structure because free Ce6 and small molecules were removed by dialysis procedure.

  1. How to identify the Ce6 contents in Table 1? The authors did not provide the information in the main text.

Answer) Thanks for your valuable comment. We measured Ce6 contents in the copolymer was as following method: To analyze Ce6 contents in the copolymer, FA-PEG3500-ss-Ce6 or FA-PEG3500-ss-Ce6tri copolymer  (5 mg) were reconstituted in 50 mL PBS in the presence of 20 mM DTT and then stirred for 24 h. This solution was diluted ten times with DMSO to measure Ce6 contents. Ce6 contents were evaluated with an Infinite M200 pro microplate reader (Tecan, Mannedorf, Switzerland) at excitation wavelength of 407 nm and emission wavelength of 664 nm. Standard test was performed with Ce6 alone in DMSO. Ce6 content, % (w/w) = (Ce6 weight/ weight of FA-PEG3500-ss-Ce6tri copolymer) × 100.

  We indicated it in the Table 1.

In Table 1. a Ce6 contents (%, w/w) = w of Ce6/w of copolymer. Ce6 contents in the copolymer were evaluated as decribed in 2.6. Ce6 release study: 5 mg of copolymer in 50 ml PBS was incubated with DTT for 24 h and then this solution was diluted with DMSO ten times.

  1. In Figure 2, the copolymer was treated with GSH to form a complex structure. The size distribution becomes larger and wider, but the TEM image only shows a collapsed structure and the size became smaller. The authors should explain this point.

Answer) Thanks for your valuable comment. Practically, when TEM samples prepared, degraded products must be dispersed in the medium and then small particles were observed evidently. As shown in Figure 2B(b), small particles were found with dirty/shadow parts. This is the reason for smaller particles and dirty images in the TEM pictures.

Furthermore, morphology of nanophotosensitizers became irregular form by treatment with GSH as shown in Figure 2B(b)) while no treatment with GSH resulted in spherical morphology (Figure 2B(a)). These results indicate that nanophotosensitizers can be disintegrated by GSH (Figire2B(b)).

  1. In Figure 3, the authors should address the reason regarding the fluorescence increment of the Ce6 released from the nanoparticles upon GSH treatment.

Answer) Thanks for your valuable comment. Practically, Ce6 may be existed at quenching state when it conjugated with polymer. However, when Ce6 was liberated from nanophotosensitizers, it must be activated and its fluorescence intensity was increased. We discussed more in the Results and discussion section.

Liberated Ce6 became activated state in the aqueous solution while Ce6 in the nanophotosensitizers are existed as a ground state and then revealed decreased fluorescence intensity as shown in Figure 3(a).

  1. The PDT efficacy of Ce6 alone and nanophotosensitizers at different concentrations were compared in Figure 5. How to determine the Ce6 concentration loaded onto the nanoparticles?

Answer) Thanks for your valuable comment. According to your comment, we indicated it in the section “2.6. Ce6 release study”. Practically, we determined Ce6 concentration in the copolymer as follows: To analyze Ce6 contents in the copolymer, FA-PEG3500-ss-Ce6 or FA-PEG3500-ss-Ce6tri copolymer  (5 mg) were reconstituted in 50 mL PBS in the presence of 20mM DTT and then stirred for 24 h. This solution was diluted ten times with DMSO to measure Ce6 contents. Ce6 contents were evaluated with an Infinite M200 pro microplate reader (Tecan, Mannedorf, Switzerland) at excitation wavelength of 407 nm and emission wavelength of 664 nm. Standard test was performed with Ce6 alone in DMSO. Ce6 content, % (w/w) = (Ce6 weight/ weight of FA-PEG3500-ss-Ce6tri copolymer) × 100.

  1. Recent articles regarding the PDT therapy of Ce6-loaded hybrid nanoparticles can be cited: for example, Wang, BY et al. Nanomaterials 2017, 7, p269; Sun X et al. J. Mater. Chem. B, 2019, 38, p5797.

Answer) Thanks for your valuable comment. According to your comment, we cited these references in the Introduction and Discussion part.

  1. Sun, X.; Sun, J.; Lv, J.; Dong, B.; Liu, M.; Liu, J.; Sun, L.; Zhang, G.; Zhang, L., Huang, G.; Xu, W.; Xu, L.; Bai, X.; Song, H. Ce6-C6-TPZ co-loaded albumin nanoparticles for synergistic combined PDT-chemotherapy of cancer. J. Mater. Chem. B. 2019, 7, 5797-5807.
  2. Wang, B.Y.; Liao, M.L.; Hong, G.C.; Chang, W.W.; Chu, C.C. Near-Infrared-Triggered Photodynamic Therapy toward Breast Cancer Cells Using Dendrimer-Functionalized Upconversion Nanoparticles. Nanomaterials 2017, 7, 269.

Reviewer 4 Report

The manuscript “Nanophotosensitizers for folate receptor targeted and redox-sensitive delivery of chlorin e6 against cancer cells” (materials-817564) reported a folate acid targeted amphiphilic macromolecular photosensitizer with double sulfide bonds in the main chain for GSH responsive delivery of hydrophobic photensitizer Ce6 for PDT of oral cancer. The system is interesting which like a prodrug, where Ce6 can be efficiently delivery to target cancer cell and then release therein for PDT. However, the paper needed a major revision before a positive decision.

  • MTT studies should be added to make sure the PDT performance of the nanophotosensitizer for the both kinds of cell.
  • In vivo studied should be added to makes it a complete research for oral cancer treatment.

Author Response

Answer to reviewer 4’s comment

The manuscript “Nanophotosensitizers for folate receptor targeted and redox-sensitive delivery of chlorin e6 against cancer cells” (materials-817564) reported a folate acid targeted amphiphilic macromolecular photosensitizer with double sulfide bonds in the main chain for GSH responsive delivery of hydrophobic photensitizer Ce6 for PDT of oral cancer. The system is interesting which like a prodrug, where Ce6 can be efficiently delivery to target cancer cell and then release therein for PDT. However, the paper needed a major revision before a positive decision.

MTT studies should be added to make sure the PDT performance of the nanophotosensitizer for the both kinds of cell.

Answer) Thanks for your valuable comment. We added MTT cell proliferation assay (as a phototoxicity and dark toxicity) in the Figure 5(a) and (b). Thanks.

In vivo studied should be added to makes it a complete research for oral cancer treatment.

Answer) Thanks for your valuable comment. At this moment, we focused on the synthesis of FA- PEG3500-ss-Ce6tri nanophotosensitizers and then approved their potential to target folate receptor with KB cells. We have plan to do in vivo anticancer study using tumor-xenograft mouse model and we are waiting for permission of animal study from our department. We will report it in the future report.

Reviewer 5 Report

The work «Nanophotosensitizers for folate receptor targeted and 3 redox-sensitive delivery of chlorin e6 against cancer cells» from Min-Suk Kook et al. describes the synthesis and characterization of FA-PEG-ss-Ce6tri copolymer to deliver photosensitizers via redox-sensitive and folate-specific manner in order to apply PDT to oral cancer cell lines.

This work is interesting, but the paper needs a careful revision in order to be suitable for publication in Materials as it appears neglected in some aspects. The English must be also improved.

The experimental part is lacking some details and some performed experiments are not clearly described.

Main remarks:

Page 4: Paragraph 2.8, “PDT treatment”, the authors should indicate the time of irradiation. It’s not clear the culture conditions after irradiation, which medium they use, with or without serum.

Pages 4 and 5, lines 187 and 188 respectively: The authors claim that they fixed the cells with “Immunomount”. In my knowledge, this compound is a mounting media not a fixative one.

Page 5: Paragraph 2.11 “ROS production”, again, the time of irradiation is not indicated.

Table 1: What it means “Ce6 contents (%, w/w)”. I suppose that it means “w of Ce6/w of copolymer”. The authors should explain this and add it under the table.

Page 8: In the legend of Figure2B, what is the corresponding compound of the concentrations (a) 0 mM; (b) 10 mM.

The authors characterized the size of their copolymer. What ‘s about their charge (zeta potential)?

Page 8, Figure 3b, what is the concentration of the added GSH?

Page 9, Figure 4: what is the length of the scale bar?

Figures 5 and 6: the legends of these figures are too short, the authors must give more details. This is true for all figures.

Page 11, Figure 6, the length of scale bar and the description of panel (d) are missing.

The authors should discuss in details the differences in PDT efficacy between Ce6 vs FA-PEG-ss-Ce6tri copolymer. They have also to discuss their results in comparison with the previous publications of:

- Donghong Li et al., Journal of Photochemistry and Photobiology B: Biology, Volume 127, 2013, pp. 28-37.

- Hyung Park Kun Na. Biomaterials, Volume 34, Issue 28, September 2013, Pages 6992-7000

Minor remarks:

Page 7, line 257: Figure instead of Figire

Page 9, line 279, microscopy instead of miscopy

Page 12, line 320: repeated sentence, “…KB cells as shown in Figure 7. As shown in Figure 7…”

Author Response

Answer to reviewer 5’s comment

The work «Nanophotosensitizers for folate receptor targeted and 3 redox-sensitive delivery of chlorin e6 against cancer cells» from Min-Suk Kook et al. describes the synthesis and characterization of FA-PEG-ss-Ce6tri copolymer to deliver photosensitizers via redox-sensitive and folate-specific manner in order to apply PDT to oral cancer cell lines.

This work is interesting, but the paper needs a careful revision in order to be suitable for publication in Materials as it appears neglected in some aspects. The English must be also improved.

Answer) Thanks for your valuable comment. We fully addressed the reviewer’s comments and revised the manuscript.

The experimental part is lacking some details and some performed experiments are not clearly described.

Answer) Thanks for your valuable comment. We revised the manuscript in experimental section. We added all of performed experiments In Materials and methods section.

Main remarks:

Page 4: Paragraph 2.8, “PDT treatment”, the authors should indicate the time of irradiation. It’s not clear the culture conditions after irradiation, which medium they use, with or without serum.

Answer) Thanks for your valuable comment. According to your comment, we indicated the time of irradiation. Irradiation time with our equipment was 9min 29s and then cells were cultured in serum-free media.

2.8. PDT effect in vitro

The dose of light irradiation was measured with a photo radiometer (Delta Ohm, Padua, Italy) and the cells were irradiated for 9 min 29 s (2.0 J/cm2). Following this, cells were further incubated with serum-free media for 1 day at 37 oC in 5 % CO2.

Pages 4 and 5, lines 187 and 188 respectively: The authors claim that they fixed the cells with “Immunomount”. In my knowledge, this compound is a mounting media not a fixative one.

Answer) Thanks for your valuable comment. According to your comment, we corrected it. 2.10. Fluorescence microscopy for observation of Cells

2.10. Fluorescence microscopy for observation of Cells

Following this, cells were washed with PBS twice and fixed with 4 % paraformaldehyde solution. They were mounted with immobilization solution (Immunomount, thermo Electron Co. Pittsburgh, PA, USA) and then observed with fluorescence microscope (Eclipse 80i; Nikon, Tokyo, Japan).

2.11. Folate receptor-targeting of KB cells

Following this, cells were washed with PBS and then fixed with 4 % paraformaldehyde solution. This was mounted with ImmuMount solution (Thermo Electron Corporation, Pittsburgh, PA). Cells were observed with fluorescence microscope.

Page 5: Paragraph 2.11 “ROS production”, again, the time of irradiation is not indicated.

Answer) Thanks for your valuable comment. According to your comment, we indicated the time of irradiation. Irradiation time with our equipment was 9min 29s.

2.12. ROS production

The cells were irradiated for 9 min 29 s (2.0J/cm2) at 664 nm using expanded homogenous beam.

Table 1: What it means “Ce6 contents (%, w/w)”. I suppose that it means “w of Ce6/w of copolymer”. The authors should explain this and add it under the table.

Answer) Thanks for your valuable comment. Your comment was exact and we corrected it in the manuscript.

Page 8: In the legend of Figure2B, what is the corresponding compound of the concentrations (a) 0 mM; (b) 10 mM.

Answer) Thanks for your valuable comment. The corresponding compound was GSH. We indicated it in the Figure.

Figure 2. A. Particle size distribution of FA-PEG3500-ss-Ce6tri copolymer nanophotosensitizers. GSH concentration: (a) GSH, 0 mM; (b) GSH, 5 mM; (c) GSH, 10 mM; (d) GSH, 20 mM. B. Morphological observations of FA-PEG3500-ss-Ce6tri copolymer nanophotosensitizers. (a) GSH, 0 mM; (b) GSH, 10 mM.

The authors characterized the size of their copolymer. What‘s about their charge (zeta potential)?

Answer) Thanks for your valuable comment. At this moment, we focused on the drug targeting to the tumor cells in this article. We will check it and report in the near future with additional characterization results.

Page 8, Figure 3b, what is the concentration of the added GSH?

Answer) Thanks for your valuable comment. The concentration of GSH was 0 mM for GSH(-) and 20 mM for GSH(+). We indicated in the Figure.

Figure 3. (a) The effect of GSH concentration on the fluorescence spectra of FA-PEG3500-ss-Ce6tri copolymer nanophotosensitizers. For fluorescence images, nanophotosensitizer solution (0.1 mg/ml) in PBS (pH 7.4, 0.01M) was incubated at 37oC for 3 h in the presence of various concentrations of GSH to analyze the effect of GSH. (b) Ce6 release from FA-PEG3500-ss-Ce6tri copolymer nanophotosensitizers in the presence or absence of GSH. GSH concentration in the media was 0 mM for GSH(-) and 20 mM for GSH(+).

Page 9, Figure 4: what is the length of the scale bar?

Answer) Thanks for your valuable comment. The magnification of photos in this Figure was 400x. We indicated the magnification in the Figure legend.

Figure 4. Ce6 uptake ratio (left panel) and fluorescence microscopic observation (right panel) of Ce6 alone and FA-PEG3500-ssCe6tri copolymer nanophotosensitizers. (a) KB cells; (b) YD-38 cells. 2×104 cells) were treated with various concentrations of Ce6 or nanophotosensitizers for 2 h. Magnification of fluorescence images was 400 x.

Figures 5 and 6: the legends of these figures are too short, the authors must give more details. This is true for all figures.

Answer) Thanks for your valuable comment. According to your comment, we fully revised the Figure Legend.

Figure 1. (a) Synthesis scheme and (b) 1H NMR spectra of FA-PEG3500-ssCe6tri copolymer. FA-PEG3500-ssCe6tri copolymer was dissolved in DMSO-d form and then measured with NMR spectroscopy.

Figure 2. A. Particle size distribution of FA-PEG3500-ss-Ce6tri copolymer nanophotosensitizers. GSH concentration: (a) GSH, 0 mM; (b) GSH, 5 mM; (c) GSH, 10 mM; (d) GSH, 20 mM. B. Morphological observations of FA-PEG3500-ss-Ce6tri copolymer nanophotosensitizers. (a) GSH, 0 mM; (b) GSH, 10 mM.

Figure 3. (a) The effect of GSH concentration on the fluorescence spectra of FA-PEG3500-ss-Ce6tri copolymer nanophotosensitizers. For fluorescence images, nanophotosensitizer solution (0.1 mg/ml) in PBS (pH 7.4, 0.01M) was incubated at 37oC for 3 h in the presence of various concentrations of GSH to analyze the effect of GSH. (b) Ce6 release from FA-PEG3500-ss-Ce6tri copolymer nanophotosensitizers in the presence or absence of GSH. GSH concentration in the media was 0 mM for GSH(-) and 20 mM for GSH(+).

Figure 4. Ce6 uptake ratio (left panel) and fluorescence microscopic observation (right panel) of Ce6 alone and FA-PEG3500-ssCe6tri copolymer nanophotosensitizers. (a) KB cells; (b) YD-38 cells. 2×104 cells) were treated with various concentrations of Ce6 or nanophotosensitizers for 2 h. Magnification of fluorescence images was 400 x.

Figure 5. PDT efficacy of FA-PEG3500-ssCe6tri copolymer nanophotosensitizers. (a) Dark toxicity; (b) Phototoxicity; (c) ROS production. Dark toxicity and phototoxicity of KB cells and YD-38 cells were evaluated with MTT proliferation assay. For phototoxicity, cells were treated with Ce6 or nanophotosensitizers for 2 h and, after that, irradiated at 664 nm for 9 min 29 s (2.0J/cm2). For dark toxicity, cells were treated ith Ce6 or nanophotosensitizers similarly in the absence of irradiation.

Figure 6. (a) Fluorescence microscopic observations of KB cells treated with Ce6 alone or FA-PEG3500-ssCe6tri copolymer nanophotosensitizers. FA was pretreated to the cells to block folate receptor of cancer cells. (b) Ce6 uptake ratio; (c) ROS production; (d) Phototoxicity. Competition assay for folate receptor-sensitivity of KB cells, cells were exposed to 5 mM folic acid for 1 h and, after that, cells were washed Ce6 alone or nanophotosensitizers (2 mg/ml as a Ce6) were treated to cells. Following procedures was similar to those of Figure 4 and 5. Magnification : 200 x

Figure 7. Pulmonary metastasis of KB cells. To block folate receptor of KB cells, folic acid in PBS (dose: 20 mg/kg) was i.v. injected 1h before injection of nanophotosensitizer. 1 × 106 KB cells were intravenously (i.v.) injected into the tail vein of nude BALb/C mouse and, three weeks later, nanophotosensitizers (dose: 10 mg/kg) were injected intravenously (i.v.) via the tail vein of the mouse. To block folate receptor of tumors (FA(+)), folic acid in PBS (dose: 20mg/kg) was i.v. injected 1h before injection of nanophotosensitizer. PBS without FA was i.v. injected for comparison (FA(-)). 24 h later, the mice were sacrificed to observe fluorescence images of each organ.

Page 11, Figure 6, the length of scale bar and the description of panel (d) are missing.

Answer) Thanks for your valuable comment. In the fluorescence images, magnification was 200x. We indicated it in the Figure legend. Panel (d) is a Phototoxicity. We corrected it.

Figure 6. (a) Fluorescence microscopic observations of KB cells treated with Ce6 alone or FA-PEG3500-ssCe6tri copolymer nanophotosensitizers. FA was pretreated to the cells to block folate receptor of cancer cells. (b) Ce6 uptake ratio; (c) ROS production; (d) Phototoxicity. Competition assay for folate receptor-sensitivity of KB cells, cells were exposed to 5 mM folic acid for 1 h and, after that, cells were washed Ce6 alone or nanophotosensitizers (2 mg/ml as a Ce6) were treated to cells. Following procedures was similar to those of Figure 4 and 5. Magnification : 200 x

The authors should discuss in details the differences in PDT efficacy between Ce6 vs FA-PEG-ss-Ce6tri copolymer. They have also to discuss their results in comparison with the previous publications of: - Donghong Li et al., Journal of Photochemistry and Photobiology B: Biology, Volume 127, 2013, pp. 28-37. - Hyung Park Kun Na. Biomaterials, Volume 34, Issue 28, September 2013, Pages 6992-7000

Answer) Thanks for your valuable comment. According to your comments, we cited these references and discussed more in Discussion part.

Li et al. also reported that photosensitizer conjugated with FA-PEG can be delivered by folate receptor-specific manner against folate-receptor-overexpressing cancer cells [31]. Especially, folate receptor expression is known to be increased in the patients having periodontal disease [32]. Alkan et al., reported that expression values of folate-receptor 1 was higher in gingivitis and periodontitis groups than those of healthy group [32].

Especially, tumor tissue-specific cytotoxicity during PDT can be triggered by irradiation of specific disease sites even though photosensitizers are fully distributed around the human body because PDT with did not reveal cellular cytotoxicity in the absence of light irradiation [39].

Li, D.; Li, P.; Lin H.; Jiang, Z.; Guo, L.; Li, B. A novel chlorin–PEG–folate conjugate with higher water solubility, lower cytotoxicity, better tumor targeting and photodynamic activity. J. Photochem. Photobiol. B. 2013, 127, 28-37.

Park, H.; Na, K. Conjugation of the Photosensitizer Chlorin e6 to Pluronic F127 for Enhanced Cellular Internalization for Photodynamic Therapy. Biomaterials. 2013, 34, 6992-7000.

 Minor remarks:

Page 7, line 257: Figure instead of Figire

Answer) Thanks for your valuable comment. We corrected Figire to Figure.

Page 9, line 279, microscopy instead of miscopy

Answer) Thanks for your valuable comment. We corrected miscopy of microscopy.

Page 12, line 320: repeated sentence, “…KB cells as shown in Figure 7. As shown in Figure 7…”

Answer) Thanks for your valuable comment. According to your comment, we removed repeated sentences, “As shown in Figure 7,”

Folate receptor targetability of FA-PEG3500-ss-Ce6tri nanophotosensitizers was also studied using pulmonary metastasis model of KB cells as shown in Figure 7. KB cells were i.v. administered to the tail vein of the mice to induce pulmonary metastasis.

Round 2

Reviewer 2 Report

The manuscript is now acceaptable for publication in "Materials"

Author Response

The manuscript is now acceaptable for publication in "Materials"

Answer) Thank you for your decision.

Reviewer 3 Report

The manuscript has been revised following all the reviewer's comments. I can recommend it to be published in the journal of "Materials".

Author Response

The manuscript has been revised following all the reviewer's comments. I can recommend it to be published in the journal of "Materials".

Answer) Thanks for your decision.

Reviewer 4 Report

By now, it can be accepted.

Author Response

By now, it can be accepted.

Answer) Thanks for your decision.

Reviewer 5 Report

The authors answered to almost the comments but still two points to be clarified before acceptation for publication:

First is the medium used for cells incubation after irradiation. In my knowledge, the use of serum-free medium is not appropriate because in this condition, the viability of the cells is not optimal. The authors should repeat this experiment in presence of serum.

Second, the scale bars are missing in somme panels of the figures 4 and 6,  and importantly the length units (µm) of the scale bars are not provided.

Author Response

The authors answered to almost the comments but still two points to be clarified before acceptation for publication:

Answer) Thank you for your valuable comment.

First is the medium used for cells incubation after irradiation. In my knowledge, the use of serum-free medium is not appropriate because in this condition, the viability of the cells is not optimal. The authors should repeat this experiment in presence of serum.

Answer) Thank you for your valuable comment. Practically, in this experiment, we studied anticancer PDT effect of nanophotosensitizers by the factor of cell cytotoxicity (with serum plus medium) rather than cell growth (serum-free medium). That is, we focused on whether or not generated ROS can directly kill the cancer cells rather than inhibition of growth of cancer cells. Then we selected the factors of cell cytotoxicity rather than growth inhibition of cells. Furthermore, the other reason is that generated ROS (by PDT using nanophotosensitizers) can be slightly scanvenged by FBS in the medium (Mun SE et al., Dual Effect of Fetal Bovine Serum on Early Development Depends on Stage-Specific Reactive Oxygen Species Demands in Pigs. Plos One, 2017;12:e0175427.). Therefore, we performed PDT study in the absence of FBS. Anyway, previous studies also reported the PDT effect of photosensitizers or mnanophotosensitizers in the absence of FBS. For example, Kim et al., also studied the PDT effect of nanophotosensitizers using serum-free medium and they also abtained positive results from these experiment. Jung et al., also used serum-free medium for PDT efficacy of nanophotosensitizers. Anyway, in the future study we will precisely investigate the scavenging effect (or not) of FBS on the ROS generation and will report it separated paper. Thanks again for your valuable comment.

Kim DM et al., CD44 Receptor-Specific and Redox-Sensitive Nanophotosensitizers of Hyaluronic Acid-Chlorin e6 Tetramer Having Diselenide Linkages for Photodynamic Treatment of Cancer Cells. J Pharm Sci. 2019 Nov;108(11):3713-3722.

Jung S. et al., Hyaluronic Acid-Conjugated With Hyperbranched Chlorin e6 Using Disulfide Linkage and Its Nanophotosensitizer for Enhanced Photodynamic Therapy of Cancer Cells. Materials (Basel). 2019;12:3080.

Second, the scale bars are missing in somme panels of the figures 4 and 6, and importantly the length units (µm) of the scale bars are not provided.

Answer) Thank you for your valuable comment. According to your comment, we revised the manuscript and corrected the Figures. The length units (µm) of the scale bars were indicated in the Figure legends.

Figure 4. Ce6 uptake ratio (left panel) and fluorescence microscopic observation (right panel) of Ce6 alone and FA-PEG3500-ssCe6tri copolymer nanophotosensitizers. (a) KB cells; (b) YD-38 cells. 2×104 cells) were treated with various concentrations of Ce6 or nanophotosensitizers for 2 h. Bar = 20 mm.

Figure 6. (a) Fluorescence microscopic observations of KB cells treated with Ce6 alone or FA-PEG3500-ssCe6tri copolymer nanophotosensitizers. FA was pretreated to the cells to block folate receptor of cancer cells. (b) Ce6 uptake ratio; (c) ROS production; (d) Phototoxicity. Competition assay for folate receptor-sensitivity of KB cells, cells were exposed to 5 mM folic acid for 1 h and, after that, cells were washed Ce6 alone or nanophotosensitizers (2 mg/ml as a Ce6) were treated to cells. Following procedures was similar to those of Figure 4 and 5. Bar = 40 mm.
